# Infrared and Visible Image Homography Estimation Using Multiscale Generative Adversarial Network

**Yinhui Luo, Xingyi Wang \*** , **Yuezhou Wu and Chang Shu**

School of Computer Science, Civil Aviation Flight University of China, Guanghan 618307, China
\* Correspondence: wangxingyi_97@163.com

**Abstract:** In computer vision, the homography estimation of infrared and visible multi-source images based on deep learning is a current research hotspot. Existing homography estimation methods ignore the feature differences of multi-source images, which leads to poor homography performance in infrared and visible image scenes. To address this issue, we designed an infrared and visible image homography estimation method using a Multi-scale Generative Adversarial Network, called HomoMGAN. First, we designed two shallow feature extraction networks to extract fine features of infrared and visible images, respectively, which extract important features in source images from two dimensions: color channel and imaging space. Second, we proposed an unsupervised generative adversarial network to predict the homography matrix directly. In our adversarial network, the generator captures meaningful features for homography estimation at different scales by using an encoder–decoder structure and further predicts the homography matrix. The discriminator recognizes the feature difference between the warped and target image. Through the adversarial game between the generator and the discriminator, the fine features of the warped image in the homography estimation process are closer to the fine features of the target image. Finally, we conduct extensive experiments in the synthetic benchmark dataset to verify the effectiveness of HomoMGAN and its components. We conduct extensive experiments and the results show that HomoMGAN outperforms existing state-of-the-art methods in the synthetic benchmark datasets both qualitatively and quantitatively.

**Keywords:** homography estimation; generative adversarial network; infrared image; visible image

## 1. Introduction

The perception of visible images can be severely impaired under certain environmental conditions. However, infrared images are less affected by illumination changes and can avoid such problems [1]. Due to infrared and visible images having good complementary properties, they have been extensively studied in image fusion tasks [2–6]. Homography estimation is used to compute the mapping relationship from one image to another and is a crucial upstream task in image fusion [7]. However, homography estimation between infrared and visible images is challenging due to their significant imaging differences. Therefore, this study focuses on the homography estimation between infrared and visible images to provide technical support for image registration and fusion tasks.

Traditional homography estimation methods usually utilize the feature points extracted from the image pair to obtain a set of feature correspondences [8–10] and then use robust estimation algorithms [11–13] to remove the outliers, which leads to obtaining the homography matrix. However, the shared features between infrared and visible image pairs are highly unstable, and it is difficult for such methods to obtain high-precision homography matrices. Due to the excellent feature-extraction ability of neural networks, they can be used to solve the challenges posed by traditional methods. Recently, unsupervised learning methods [14–17] have gained popularity in homography estimation. These methods optimize the model by minimizing the distance from the warped image to the

target image and have good performance in relation to homologous image pairs. Some methods [15–17] use a three-layer convolution with shared weights to extract the shallow features of image pairs and use them for loss calculation. However, the shared weights in the network optimization process did not consider the feature differences in multi-source images for infrared and visible images, resulting in the introduction of significant noise in the feature extraction process. In addition, due to the grayscale difference between infrared and visible images, most deep learning-based methods easily fail to converge.

Most deep learning-based methods are prone to poor performance due to the difference in the characteristics of infrared and visible images, but the self-optimization ability of the Generative Adversarial Network (GAN) can effectively solve such problems. The GAN makes the fine-feature map of the warped image closer and closer to the fine-feature map of the target image, thereby forcing the homography matrix to be more accurate. However, most homography estimation methods based on deep learning rarely consider the use of the GAN for self-optimization, i.e., the accuracy of the homography matrix is continuously improved by the confrontation game process between the generator and the discriminator. In particular, Hong et al. [17] utilized the GAN to impose coplanar constraints on the predicted homography by using a generator to predict the masks of aligned regions. However, the self-optimization object of this method is the mask, and a two-stage network training strategy is used, i.e., the mask is introduced to optimize it when the predicted homography matrix is guaranteed to be relatively accurate. This not only makes the training process more complicated but also does not directly self-optimize the homography matrix itself. This naturally raises the question: is it possible to use the GAN to predict the output homography matrix and self-optimize it directly?

To address the problem of poor homography estimation performance caused by existing methods ignoring feature differences in multi-source images, this study proposes an infrared and visible image homography estimation method using multi-scale generative adversarial network (HomoMGAN). This method describes the homography estimation as an adversarial game process to achieve high-precision mapping between the fine features of the warped and target image. HomoMGAN consists of two shallow feature extraction networks (infrared shallow feature extraction network and visible shallow feature extraction network) and a GAN (generator and discriminator). First, we employ the infrared shallow feature extraction network and the visible shallow feature extraction network to extract fine-feature maps of infrared and visible images to reduce the noise introduced due to the feature differences, respectively. Second, we use the GAN to self-optimize the homography matrix to reduce the impact of feature differences on homography estimation. We predicted the homography matrix by channel-concatenating the fine-feature maps of infrared and visible images and feeding them into a homography estimation generator. The homography estimation generator can capture features of different scales, and then improve the homography estimation performance by fusing shallow low-level features. We also introduce a discriminator to distinguish the fine features of the warped and target images to further optimize the homography estimation. The goal of the discriminator is to force the fine-feature map of the warped image to be aligned with the target image as much as possible, thereby improving the accuracy of the homography matrix. Extensive experimental results show that our method significantly outperforms existing methods and demonstrates the effectiveness of our proposed components.

The main contributions in this paper are summarized as follows:

- We propose a shallow feature extraction network with unshared weights to extract fine-feature maps of infrared and visible images. In particular, the discriminator takes a fine-feature map produced using a shallow feature extraction network as its input.
- To the best of our knowledge, HomoMGAN is the first work to use GANs to predict the output homography matrix and self-optimize the homography matrix by an adversarial game process directly.

- This study designed a homography estimation generator that extracts multi-scale features and captures meaningful features for homography estimation at different scales by using an encoder–decoder structure.

For infrared and visible scenes, the feature maps obtained in the proposed method fully consider the differences in multi-source images and can effectively reduce the noise of feature maps, as shown in Figure 1. Additionally, the rest of the paper is organized as follows. Section 2 describes the current developments related to feature-based and deep-learning-based homography estimation methods, respectively. At the same time, the homography estimation method using GANs is introduced, and the differences from the proposed method are explained. The components and loss function of HomoMGAN are described in detail in Section 3. In Section 4, the experimental details are introduced and analyzed in detail from both qualitative and quantitative perspectives. Meanwhile, we also conduct ablation experiments to demonstrate the effectiveness of the components. Section 5 presents some conclusions and proposes future work.

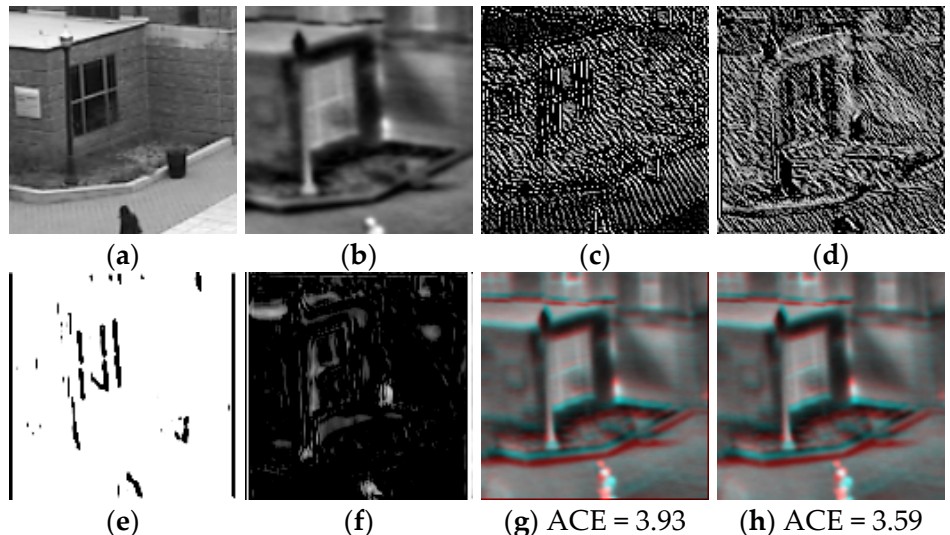

**Figure 1.** Homography estimation with the proposed method and a deep learning-based method (CADHN [15]). (**a**,**b**) represent the visible and infrared image patches, respectively. (**c**,**d**) represent the visible and infrared image feature maps obtained by the feature extractor and mask predictor in CADHN [15], respectively. (**e**,**f**) represent the visible and infrared image feature maps obtained by the shallow feature extraction network in the proposed method, respectively. (**g**,**h**) represents the channel mixing visualization method of CADHN [15] and the proposed method, respectively. The channel mixing approach is to mix the blue and green channels of the infrared warped image with the red channel of the infrared ground-truth image. The channel mixing results are mixed using uncropped image pairs of size $150 \times 150$, so the channel mixing images have more detailed information than visible and infrared image patches. Specifically, misaligned pixels appear as yellow, blue, red, or green ghosts, and the channel mixing visualization in the rest of this paper adopts this method.

## 2. Related Work

**Feature-based homography estimation.** These methods usually estimate local feature points using feature extraction algorithms, such as Scale Invariant Feature Transform (SIFT) [8], Speeded Up Robust Features (SURF) [9], Oriented FAST and Rotated BRIEF (ORB) [10], Binary Robust Invariant Scalable Keypoints (BRISK) [18], Accelerated-KAZE (AKAZE) [19], KAZE [20], Locality Preserving Matching (LPM) [21], Grid-Based Motion Statistics (GMS) [22], Boosted Efficient Binary Local Image Descriptor (BEBLID) [23], Learned Invariant Feature Transform (LIFT) [24], SuperPoint [25], Second-Order Similarity Network (SOSNet) [26], and Order-Aware Networks (OAN) [27], etc. Then, they match the feature points between the two images and use robust estimation algorithms to remove outliers, such as Random Sample Consensus (RANSAC) [11], Marginalizing Sample Con-

sensus (MAGSAC) [12], MAGSAC++ [13], Antileakage LSSA (ALLSSA) [28], etc. Finally, the homography matrix is solved using Direct Linear Transformation (DLT) [29]. However, the performance of such algorithms depends on the quality of feature points, and they often fail in infrared and visible scenarios.

**Deep homography estimation.** Recently, there have been many pioneering works regarding deep homography estimation, which can be divided into supervised and unsupervised methods. Supervised homography estimation methods mainly employ synthetic examples with ground-truth labels to train the network. DeTone et al. [30] used a Visual Geometry Group (VGG) network as the backbone to estimate the homography matrix between a pair of images, which is more robust than traditional feature-based methods. Le et al. [31] designed a dynamic-aware homography network by integrating a dynamic mask network into a multi-scale network for simultaneous homography and dynamic estimation. Shao et al. [32] proposed a local transformer network embedded in a multi-scale structure to learn the correspondence between input images of different resolutions. Unsupervised homography estimation methods mainly work by minimizing the loss between two images and warping the source image to the target image using a Spatial Transformation Network (STN) [33]. Nguyen et al. [14] proposed an unsupervised learning method to estimate the homography matrix, but this method is difficult to fit in infrared and visible scenarios. Zhang et al. [15] utilized a weight-shared feature extractor to extract the features of image pairs and learn an outlier mask to select only reliable regions for homography estimation. Ye et al. [16] proposed a homography flow representation and a low-rank representation to learn more stable features while reducing the feature rank. However, for multi-source images with large pixel differences, such as infrared and visible images, the change in homography flow is not stable enough, and the network is difficult to converge. Luo et al. [34] proposed a detail-aware deep homography estimation network to preserve more detailed information in infrared and visible images. However, the shallow feature extraction network of this method uses shared weights, ignoring the differences between multi-source images. Debaque et al. [35] proposed a supervised and unsupervised homography model for the registration of infrared and visible images.

**GAN-based homography estimation.** At present, there are few homography estimation methods based on GANs, and they are still in their infancy. Hong et al. [17] exploited an unsupervised GAN to impose co-planar constraints on the predicted homography. They solve the problem of plane-induced errors and focus the homography matrix estimation on the principal plane. However, this method does not use a GAN to directly predict the output homography matrix but only as a mask component to guide the homography matrix.

**Discussions.** Compared with Luo et al. [34], the shallow feature extraction network in our method is simpler, only consisting of a Convolutional Block Attention Module (CBAM) [36] and convolutional layers. At the same time, our shallow feature extraction network no longer shares weights. This allows our method to be less computationally accounted and consider the feature differences in multi-source images to reduce noise. Compared with Hong et al. [17], our method uses a GAN to self-optimize the homography matrix directly instead of self-optimizing the mask; this makes the training strategy of our method simpler.

## 3. Method

In this section, we first introduce the framework of HomoMGAN and then discuss the structure of the shallow feature extraction network, generator, and discriminator. Finally, this section details the generator and discriminator loss functions, respectively.

### 3.1. Overview

This section introduces a HomoMGAN used for the homography estimation of infrared and visible images. The network architecture of HomoMGAN is shown in Figure 2. It consists of four modules: two shallow feature extraction networks (infrared shallow feature extraction network $f_r(\cdot)$ and visible shallow feature extraction network $f_v(\cdot)$), a generator,

and a discriminator. Specifically, a pair of infrared and visible grayscale image patches, $I_r$ and $I_v$, with a size of $H \times W \times 1$, are provided as the input of the network. First, the infrared shallow feature extraction network $f_r(\cdot)$ and the visible shallow feature extraction network $f_v(\cdot)$ were used to extract the fine-feature maps of $F_r$ and $F_v$ of the infrared image $I_r$ and the visible image $I_v$, respectively. Second, we concatenate two fine-feature maps in the channel dimension and feed them into the generator to generate the homography matrix. Then, we obtain the corresponding warped images of $I_v'$ and $I_r'$ through the homography matrices of $H_{vr}$ and $H_{rv}$, respectively, and use the corresponding shallow feature extraction network to generate fine-feature maps $F_v'$ and $F_r'$. Finally, we feed $F_v'$ and $F_r$ and $F_r'$ and $F_v$ into the discriminator to distinguish the warped images from the target images, respectively. The proposed HomoMGAN establishes an adversarial game between the generator and the discriminator, such that the fine-feature maps of the warped image grow closer and closer to the fine-feature maps in the target image. Once the generator produces warped images that the discriminator cannot distinguish during the training phase, we have achieved the expected homography matrix. Algorithm 1 shows some training details of HomoMGAN.

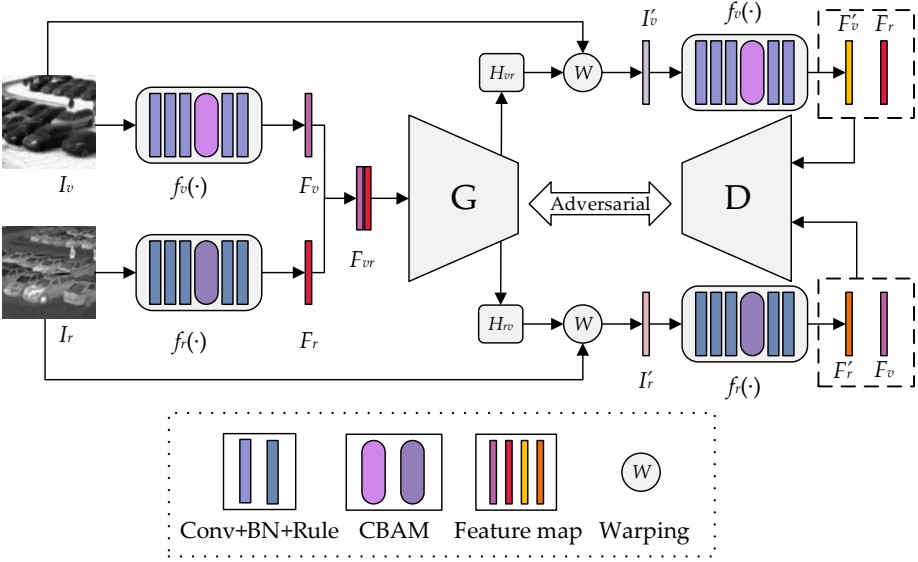

**Figure 2.** The architecture of the proposed HomoMGAN for homography estimation of infrared and visible images. Our network architecture consists of four modules: a shallow feature extraction network (infrared shallow feature extraction network $f_r(\cdot)$ and visible shallow feature extraction network $f_v(\cdot)$), a generator, and a discriminator.

---

**Algorithm 1:** The training process of HomoMGAN

**Input:** Infrared and Visible patches
**Output:** Homography matrix
**for** *number of training iterations* **do**
    **for** *k steps* **do**
        Select *m* infrared fine-feature maps $\{F_r^{(1)}, \cdots, F_r^{(m)}\}$;
        Select *m* warped visible fine-feature maps $\{F_v'^{(1)}, \cdots, F_v'^{(m)}\}$;
        Calculate the loss: $L_D(F_r, F_v')$;
        Select *m* visible fine-feature maps $\{F_v^{(1)}, \cdots, F_v^{(m)}\}$;
        Select *m* warped infrared fine-feature maps $\{F_r'^{(1)}, \cdots, F_r'^{(m)}\}$;
        Calculate the loss: $L_D(F_v, F_r')$;
        Update discriminator by AdamOptimizer: $\nabla_{\theta_D}(L_D(F_r, F_v') + L_D(F_v, F_r'))$;
    **end**
    Select *m* infrared patches $\{I_r^{(1)}, \cdots I_r^{(m)}\}$ and *m* visible patches $\{I_v^{(1)}, \cdots I_v^{(m)}\}$ from training data;
    Update generator by AdamOptimizer: $\nabla_{\theta_G} L_G$;
**end**

---

*3.2. Network Architecture*

3.2.1. Shallow Feature Extraction Network $f_v(\cdot)$ and $f_r(\cdot)$

The study designed a shallow feature extraction network to obtain fine-feature maps of images, which are used as inputs for the generator and discriminator. The shallow feature extraction network mainly consists of convolutional layers and CBAM [36]. It extracts meaningful features for homography estimation from two dimensions: channel and space. The details of its network structure are shown in Table 1. Each Conv is followed by batch normalization [37] and Rectified Linear Unit (ReLu). The infrared and visible images are captured using different sensors and have different modalities. If the network with shared weights is used for infrared and visible images to extract shallow features, their characteristics are lost, achieving a compromised result and reducing the accuracy of the homography estimation to a certain extent. Therefore, unlike the shared weights of the feature extractor and mask predictor in [15], this study designed two shallow feature extraction networks of unshared weights (infrared shallow feature extraction network $f_r(\cdot)$ and visible shallow feature extraction network $f_v(\cdot)$). In particular, the two shallow feature extraction networks have the same network structure. We use $f_v(\cdot)$ and $f_r(\cdot)$ to denote the whole process as follows:

$$F_\beta = f_\beta(I_\beta), \ \beta \in \{v, r\} \tag{1}$$

where $I_\beta$ represents the network input grayscale image.

**Table 1.** Details of shallow feature extraction network.

| Layer | Type | Kernel | Stride | Channel |
|-------|------|--------|--------|---------|
| L1 | Conv | 3 | 1 | 8 |
| L2 | Conv | 3 | 1 | 16 |
| L3 | Conv | 3 | 1 | 32 |
| L4 | CBAM | | | |
| L5 | Conv | 3 | 1 | 16 |
| L6 | Conv | 3 | 1 | 1 |

3.2.2. Generator

For the problem that most existing homography estimation networks ignore the fusion of shallow low-level features in deep networks, this study proposes a homography estimation generator to extract multi-scale features. According to our practice, these shallow low-level features can effectively improve the performance of homography estimation; a detailed description is provided in Section 4.4. Given a pair of fine-feature maps, $F_v$ and $F_r$, for channel concatenation, this study designed a generator to predict the homography matrix of the image pair. Similar to the framework of Unet [38], the generator is designed using an encoder–decoder structure to predict homography by fusing features of different scales. The network framework is shown in Figure 3. At the same time, the U-shaped structure in the generator can effectively fuse shallow low-level features in the deep network, and these shallow low-level features be beneficial to the homography estimation of image pairs.

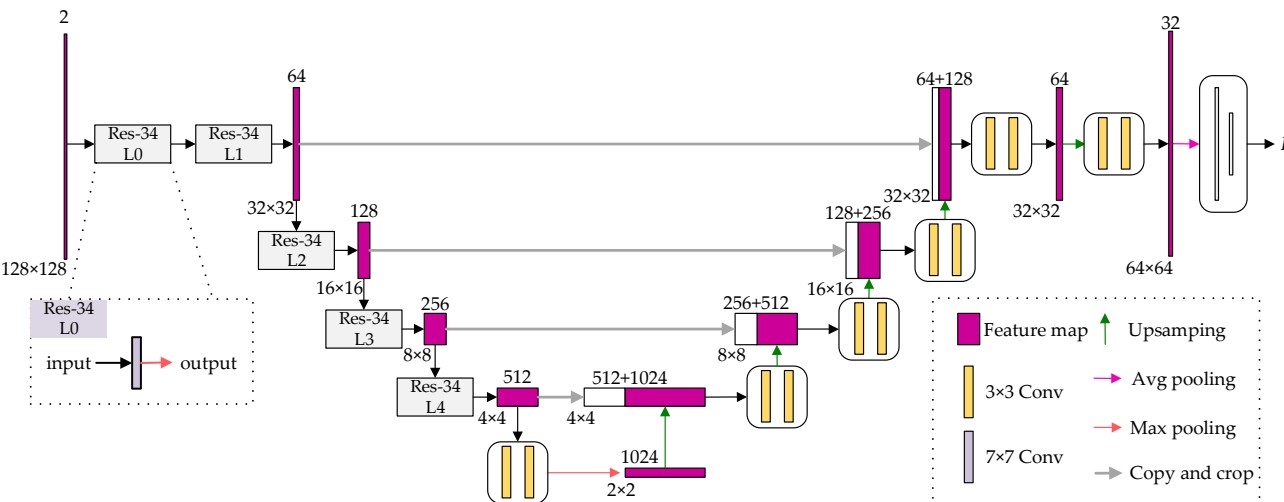

**Figure 3.** The network architecture of the generator. The network architecture of the generator is similar to the Unet [38] network, which is an encoder-decoder structure. The number in the lower left corner of the feature map in the figure represents the size of the feature map. The number above the feature map represents the number of channels on the feature map. It is worth noting that each Conv in the figure is followed by a batch normalization [37] and ReLu.

**Encoder.** This study adopted a 34-layer deep Residual Network (ResNet-34) [39] as the backbone of the encoder to obtain deeper semantic information in fine-feature maps and effectively avoid gradient vanishing and network degradation problems. It is worth noting that the average pooling and fully connected layers were removed from ResNet-34 [39] in the encoder, and the removed part was added to the end of the decoder to predict four 2D offset vectors (8 values) and use DLT [29] to obtain the homography matrix. The role of the encoder is to obtain shallow and deep semantic information of different sizes and channel numbers. Specifically, we first channel-concatenated the fine-feature maps of $F_v$ and $F_r$ as the input of the encoder. Then, the feature maps of 64 channels were obtained by the $7 \times 7$ convolution in ResNet-34 [39], and the feature maps of different scales were obtained by the four layers of ResNet-34 [39] in turn so that the encoder gets shallow and deep semantic information at different scales and channel numbers. Finally, in order to better connect the encoder with the decoder based on Unet [38], we obtained a feature map with 1024 channels by two $3 \times 3$ convolutions and max pooling.

**Decoder.** Utilizing the design idea of Unet [38], this study used the decoder in Unet [38] as the decoder in the proposed method to restore the resolution of the image by upsampling and feature copy stitching. Specifically, in each of the first four decodings, the upsampled feature maps were fused with the channel features passed from the upper-level encoder, which makes the feature maps absorb shallow low-level semantic information and obtain semantic features at different scales. Second, this study utilized a decoding module that does not fuse encoder channel features for decoding to obtain higher-resolution feature maps. Finally, the feature maps were passed through the average pooling and fully connected layers in ResNet-34 [39] to obtain four 2D offset vectors to achieve the homography matrix of the image pair.

### 3.2.3. Discriminator

Inspired by the discriminator in [40], it was utilized as the discriminator in our network, and the network architecture is shown in Figure 4. Our discriminator architecture is mainly divided into five parts. The first four parts are composed of two $3 \times 3$ convolution layers, a batch normalization layer [37], and a leaky ReLU activation function. The fifth part consists of a pooling layer and a $1 \times 1$ convolutional layer composition. Different from [40], the image pair was no longer used as the input of the discriminator, but the fine-feature map of the image pair was used as the input, which can make the discriminator take the important

feature as the main condition for the judgment, thereby reducing the unimportant influence of the characteristics. A discriminator is essentially a classifier that aims to distinguish warped images from target images. By imposing constraints on the fine-feature map through an adversarial game between the generator and the discriminator, the fine-feature map of the warped image is forced to be closer and closer to the fine-feature map of the target image, thereby improving the homography estimation performance. Once the generator generates samples that the discriminator cannot distinguish during the training phase, we achieve a relatively accurate homography matrix.

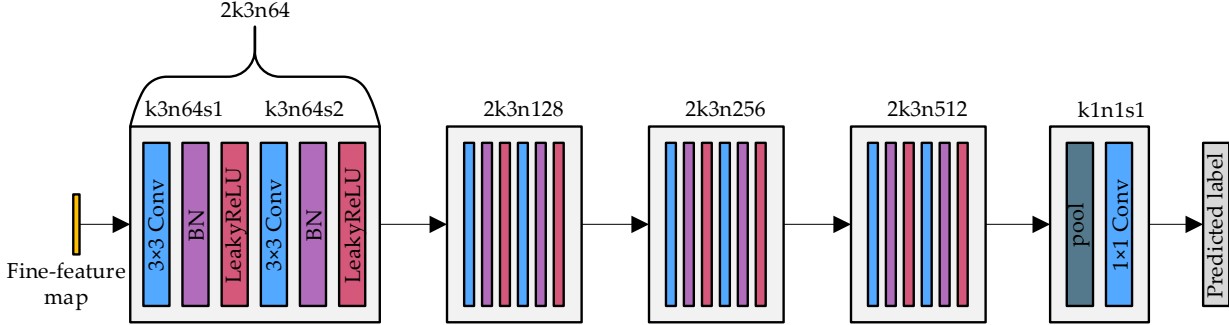

**Figure 4.** The network architecture of the discriminator. The discriminator network is similar to the VGG11 [41] network, using fine-feature maps as network input.

### 3.3. Loss Function of Generator

The loss function of the generator consists of three parts: feature loss, homography loss, and adversarial loss.

$$L_G = L_f\left(I'_r, I_v\right) + L_f\left(I'_v, I_r\right) + \lambda L_{hom} + \mu\left(L_{adv}\left(F'_r\right)\right) + L_{adv}\left(F'_v\right)\right)) \tag{2}$$

where $L_G$ represents the generator total loss. The first term $L_f\left(I'_r, I_v\right)$ and the second term $L_f\left(I'_v, I_r\right)$ on the right depict the feature loss between the warped image and the target image. The third term $L_{hom}$ represents the homography loss. The fourth term $L_{adv}\left(G, F'_r\right)$ and the fifth term $L_{adv}\left(G, F'_v\right)$ represent the adversarial loss. $\lambda$ and $\mu$ denote the factors that control the balance among the three terms. Through the analysis in Appendix A, we set $\lambda = 0.01$ and $\mu = 0.005$ in the experiment.

#### 3.3.1. Feature Loss

Since the attention mechanism is included in our shallow feature extraction network, the fine-feature maps of image pairs were used to participate in the operation directly similar to the triplet loss in [15]. The feature loss encourages the warped image fine-feature map to have a similar data distribution to the target image fine-feature map. The feature loss between the warped infrared fine-feature maps and visible fine-feature maps can be defined as follows:

$$L_f\left(I'_r, I_v\right) = \|F'_r - F_v\|_1 - \|F_r - F_v\|_1 \tag{3}$$

where $I_r$ and $I_v$ represent an infrared and visible image, respectively. $I'_r$ denotes the warped image after transforming $I_r$ using the homography matrix $H_{rv}$. $F_r$ represents the fine-feature map obtained by $I_r$ after passing through the infrared shallow feature extraction network. $F_v$ depicts the fine feature-map obtained after $I_v$ passes through the visible shallow feature extraction network. $F'_r$ represents the fine-feature map obtained after $I'_r$ passes through the infrared shallow feature extraction network. According to Equation (3), we also involved another feature loss $L_f\left(I'_v, I_r\right)$ between $I'_v$ and $I_r$.

### 3.3.2. Homography Loss

A constraint to force $H_{rv}$ and $H_{vr}$ was added to ensure they were inverse of each other, i.e.,

$$L_{hom} = \|H_{vr}H_{rv} - E\|_2^2 \tag{4}$$

where $H_{vr}$ represents the homography matrix solved by the channel concatenation of the fine-feature maps of $I_r$ and $I_v$ and sent to the generator. $H_{rv}$ represents the homography matrix solved by exchanging the fine-feature maps of $I_r$ and $I_v$ and sending them into the generator. $E$ represents a third-order identity matrix.

### 3.3.3. Adversarial Loss

The adversarial loss is defined based on the probabilities of the discriminator in all training samples, and its purpose is to force the fine-feature map of the warped image to be closer to the fine-feature map of the target image. Similarly, this study followed the idea of [40] that the adversarial loss for warped infrared fine-feature maps could be defined as follows:

$$L_{adv}(F_r') = \sum_{n=1}^{N} \left(1 - logD_{\theta_D}(F_r')\right) \tag{5}$$

where $logD_{\theta_D}(F_r')$ indicates the probability that the warped infrared fine-feature map is similar to the visible fine-feature map, and $N$ indicates the size of the batch. Similarly, another adversarial loss $L_{adv}(G_v')$ indicates that a warped visible fine-feature map can be obtained.

### 3.4. Loss Function of Discriminator

The discriminator aims to distinguish fine-feature maps between the warped and target images. The loss function of the discriminator can be defined as follows:

$$L_D = L_D(F_r, F_v') + L_D(F_v, F_r') \tag{6}$$

where $L_D$ denotes the discriminator's total loss. Both the first term $L_D(F_r, F_v')$ and the second term $L_D(F_v, F_r')$ on the right represent the loss between the fine-feature map of the warped image and the target image.

We similarly followed the idea of [40] in that the loss between the infrared fine-feature map and the warped visible fine-feature map can be defined as follows:

$$L_D(F_r, F_v') = \sum_{n=1}^{N} \left(a - logD_{\theta_D}(F_r)\right) + \sum_{n=1}^{N} \left(b - logD_{\theta_D}(F_v')\right) \tag{7}$$

where $a$ and $b$ denote the labels of the fine-feature maps $F_r$ and $F_v'$, respectively. Similarly, the loss $L_D(F_v, F_r')$ between another visible fine-feature map and the warped infrared fine-feature map can be obtained. Through the analysis in Appendix A, label $a$ is set as a random number from 0.95 to 1, and label $b$ is set as a random number from 0 to 0.05 in the experiment. The labels $a$ and $b$ are not specific numerical values but so-called soft labels [42].

## 4. Experimental Results

In this section, the dataset and implementation details are introduced, and then some experimental procedures of our method are introduced. Second, quantitative and qualitative comparisons were conducted using traditional feature-based and deep learning-based methods on a synthetic benchmark dataset to demonstrate the performance of our method. Traditional feature-based methods include nine methods consisting of five feature descriptors (SIFT [8]/ORB [10]/BRISK [18]/AKAZE [19]/KAZE [20]) and two outlier rejection algorithms (RANSAC [11]/MAGSAC++ [13]) combined. In particular, the KAZE [20] + MAGSAC++ [13] algorithm has difficulty in obtaining homography matri-

ces in infrared and visible scenes, so it was not used as a comparison method. Deep-learning-based methods include UDHN [14], CADHN [15], MBL-UDHEN [16], and HomoGAN [17]. In particular, UDHN [14] is difficult to fit in infrared and visible scenarios. MBL-UDHEN [16] and HomoGAN [17] both use the idea of homography flow, but the large grayscale and contrast differences in the infrared and visible images themselves cause the homography flow to become unstable, making it difficult for the network to converge. Therefore, we only make a comparison with CADHN [15] in the subsequent qualitative comparison and quantitative comparisons. Finally, analytical results from some ablation experiments are provided to demonstrate the effectiveness of shallow feature extraction networks and generator backbones with unshared weights.

### 4.1. Dataset and Implementation Details

4.1.1. Dataset

Considering the lack of dedicated datasets for this task, we constructed a synthetic benchmark dataset for the evaluation of homography estimation. Given the small amount of data in the registered infrared and visible datasets in the current image fusion field, we selected three classic datasets to construct an unregistered synthetic benchmark dataset: OSU Color-Thermal Database [43], INO [44], and TNO [45]. We selected 115 pairs of images for training set production and 42 pairs of images for test set production.

4.1.2. Implementation Details

Our network was implemented in PyTorch and trained on an NVIDIA GeForce RTX 3090. The network was trained using the Adam optimizer [46] with an initial learning rate of $1 \times 10^{-3}$ and a decay factor of 0.8 per epoch. Meanwhile, the batch size of the experiments was set to 48, and the epoch was set to 50. It is worth noting that the "number of training iterations" in Algorithm 1 corresponds to the epoch, and "k steps" means the quotient of the total number of training sets divided by the batch size. In addition, we chose the evaluation metrics used in typical homography estimation tasks for evaluation to objectively evaluate the performance of the homography estimation. In particular, since the corner coordinates transformed by the ground-truth homography can be obtained during the production of the synthetic benchmark dataset, we used corner error [31,32] as the evaluation index for the synthetic benchmark dataset. The corner error [31] was obtained by the average $l_2$ distance between the corner points transformed by the estimated homography and the ground-truth homography.

### 4.2. Experiment Procedure

In this section, the data preprocessing process is introduced in detail to clearly illustrate the method of constructing the synthetic benchmark dataset. Second, the feature maps and channel mixing results of image pairs during the experiment are shown to illustrate the effectiveness of our method.

4.2.1. Data Preprocessing

**Data Augmentation.** Data augmentation techniques can increase the diversity of datasets without generating new spectral or topological information [47]. Since the training set is too small to train a good model, data augmentation methods were used to expand the amount of data, such as rotation, offset, and shear. In the end, 49,738 pairs of infrared and visible images were obtained. In addition, the infrared and visible image pairs in the augmented dataset are registered, so we used the dataset production method in [30] to generate unregistered visible image $I_a$, infrared image $I_b$, and infrared ground-truth image $I_{GT}$, the production process is shown in Figure 5. Specifically, the infrared ground-truth image $I_{GT}$ is only used for the display of channel mixing results in qualitative comparison. The channel mixture of the infrared ground-truth image $I_{GT}$ and the warped image were visualized, making it easier to observe the misregistered regions and fully evaluate the performance of our method. In addition, we maintained the corner coordinates before and

after the ground-truth homography transformation in the test set production process for the calculation of corner error [31] to evaluate the homography estimation performance. It is worth noting that the size of the original infrared and visible images was inconsistent, so we uniformly upsampled or downsampled it to $150 \times 150$ in the process of making the dataset, making the image blurry. Figure 6 shows some examples from the final synthetic benchmark dataset.

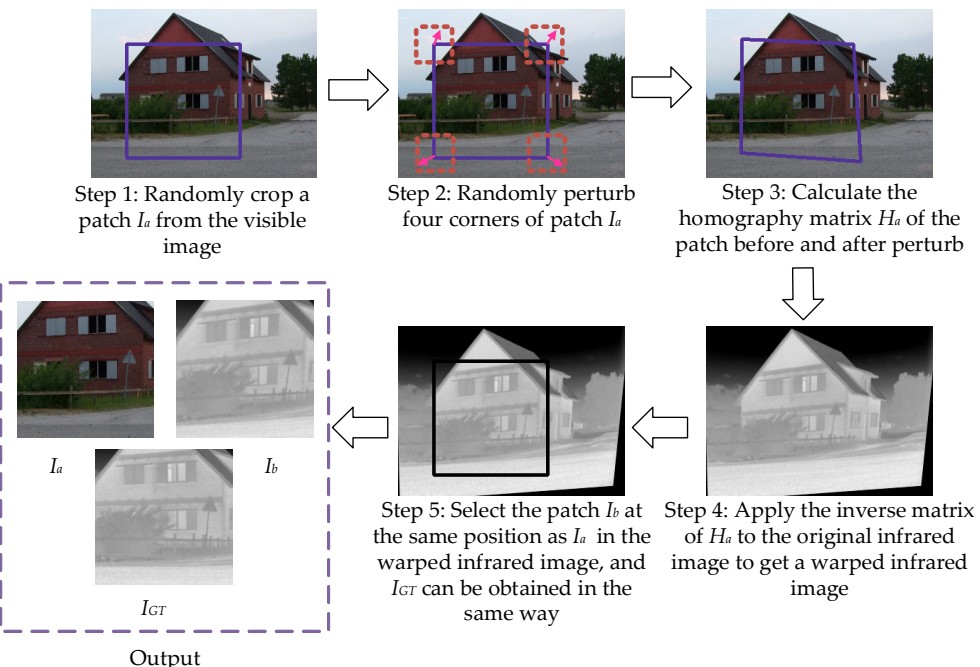

**Figure 5.** Production process of synthetic benchmark dataset.

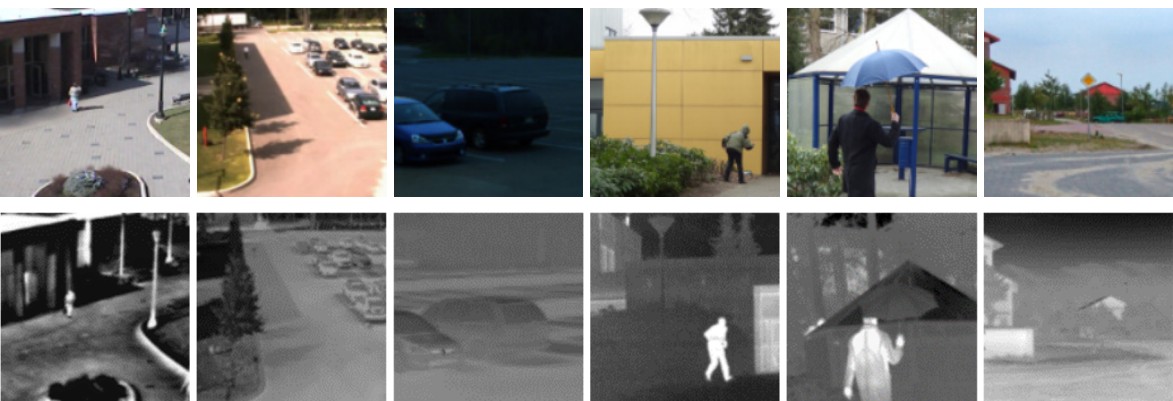

**Figure 6.** Some samples in our synthetic benchmark dataset. Row 1 represents the visible images in the synthetic benchmark dataset. Row 2 represents the infrared images in the synthetic benchmark dataset.

**Data normalization.** To allow the network to converge faster, the visible image $I_a$ and infrared image $I_b$ were normalized and grayscaled. In addition, patches with a size of $128 \times 128$ were randomly cropped from the image pair with a size of $150 \times 150$ to generate the input grayscale image pair $I_v$ and $I_r$ of the network to increase the training data.

### 4.2.2. Feature Maps and Channel Mixture Results

To demonstrate the effectiveness of our method, Figure 7 shows some fine-feature maps and channel mixing results of our method. Columns 1 and 2 show the visible image $I_v$ and its fine-feature map $F_v$ (extracted by the visible shallow feature extraction

network), respectively. Columns 3 and 4 show the infrared image $I_r$ and its fine-feature map $F_r$ (extracted by the infrared shallow feature extraction network), respectively. Column 5 shows the channel mixing results between the infrared warped image and the ground-truth image as predicted by HomoMGAN. We used two shallow feature extraction networks for infrared and visible images, enabling the network to extract feature points according to image characteristics, as shown in columns 2 and 3 in Figure 7. We can see that the features extracted by the shallow feature extraction network are relatively sparse and contain common features. Compared with dense features, sparse features reduce the impact of noise. As can be seen from the channel mixing visualization results, most regions of the image pairs are registered, which also demonstrates the effectiveness of our method.

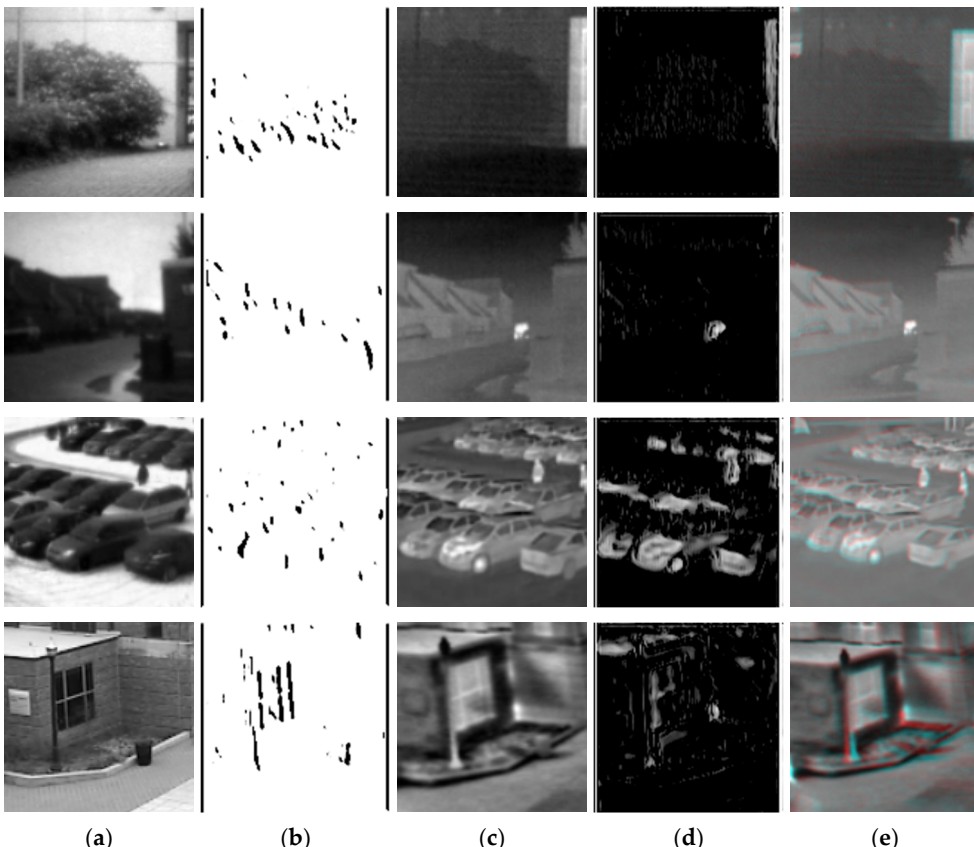

| (a) | (b) | (c) | (d) | (e) |

**Figure 7.** Fine-feature maps and channel mixing visualization results of our HomoMGAN. From left to right: (**a**) visible image patch $I_v$, (**b**) visible fine-feature map $F_v$, (**c**) infrared image $I_r$, (**d**) infrared fine-feature map $F_r$, and (**e**) channel mixing result between the warped infrared and the ground-truth image. It is worth noting that the warped image is obtained by transforming the infrared image according to the homography matrix predicted by our method.

### 4.3. Comparison on Synthetic Benchmark Dataset

**Qualitative comparison.** Qualitative comparisons were performed with ten contrasting methods in the synthetic benchmark dataset, including feature-based methods and deep learning-based methods. Figure 8 shows the comparison of the warped images obtained by using different methods based on the homography matrix, where "-" indicates that the algorithm failed. It can be seen that the feature-based methods have serious distortion and algorithm failure compared to the deep learning-based methods. In particular, SIFT [8] and AKAZE [19] appear to experience algorithm failures under these two examples. The deep learning-based methods have no distortion, and it is difficult to see the obvious difference between CADHN [15] and the proposed method.

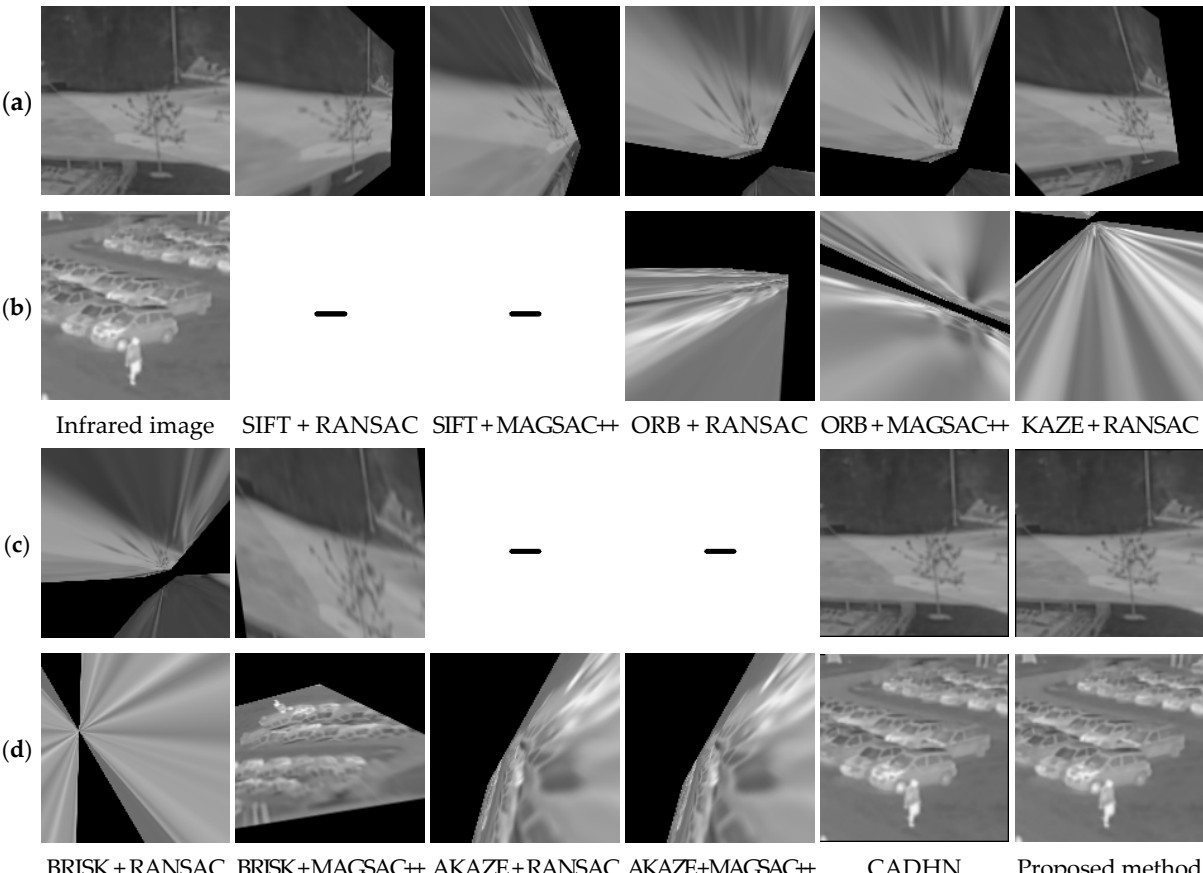

**Figure 8.** Comparison of warped images obtained by different methods in the two examples, as shown in (**a**–**d**). Except for CADHN [15], the other methods are feature-based methods.

To distinguish the unregistered region between the warped and the ground-truth image clearly, channel mixing was performed on them to more intuitively reflect the homography estimation performance, as shown in Figure 9. It can be seen that our method significantly outperforms the rest of the comparison methods. Specifically, in Figure 9a,c, SIFT [8] + RANSAC [11] is the best-performing method among the feature-based methods but slightly worse than our method. In particular, there are more unaligned regions in the deep learning-based method CADHN [15] than SIFT [8] + RANSAC [11], but CADHN [15] does not appear to exhibit algorithm failure, so feature-based methods are difficult to apply to actual scenarios. The failure of the algorithm can be observed in the quantitative comparison. For example, the curve of the method SIFT [8] + RANSAC [11] is not smooth enough and has an obvious ladder shape, as seen in Figure 10. For Figure 9b,d, the deep learning-based methods are better than the feature-based methods. All feature-based methods suffer from significant ghosting, i.e., most of the regions are misaligned. Although it is difficult to distinguish the proposed method from CADHN [15] in the channel mixing results, the superiority of the proposed method can be observed in subsequent quantitative comparisons.

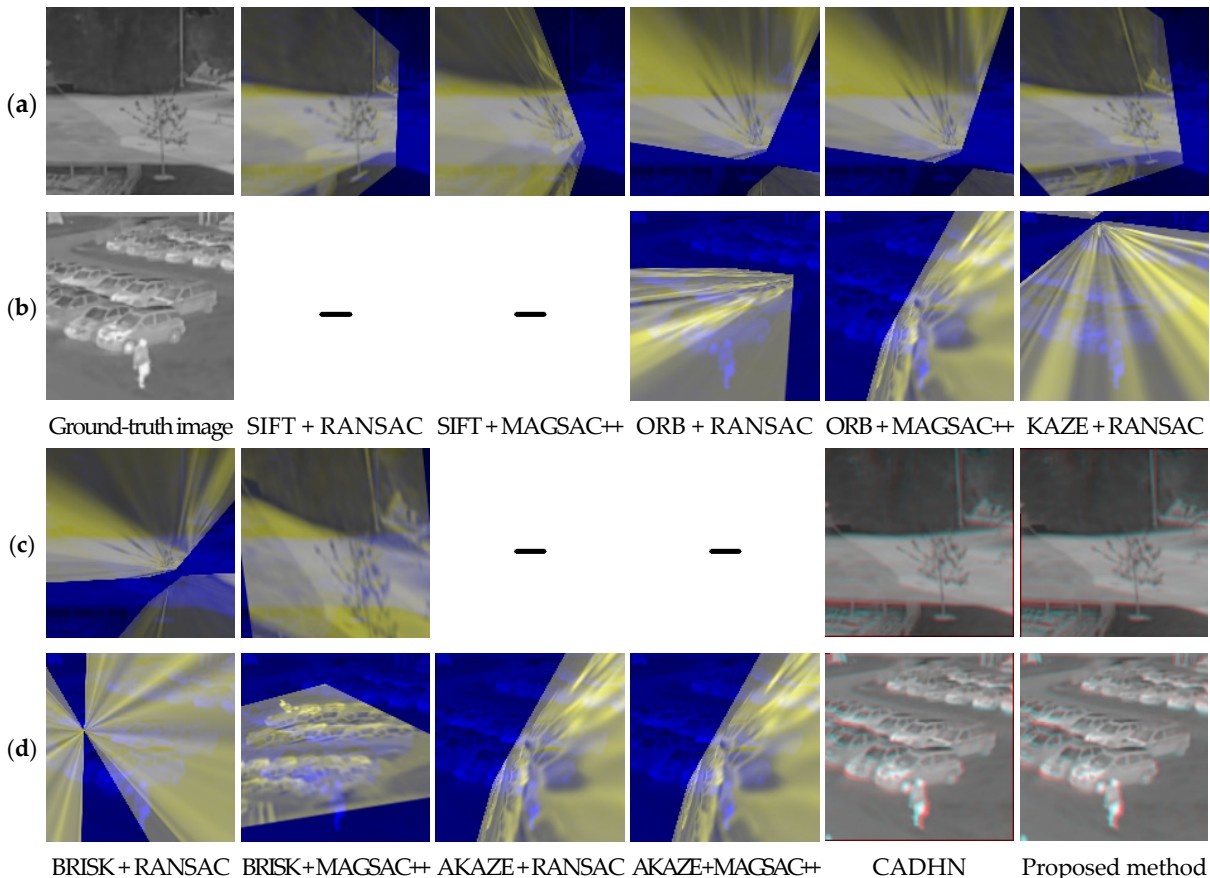

**Figure 9.** Comparison of channel mixing results of warped images and ground-truth images in two examples, as shown in (**a**–**d**).

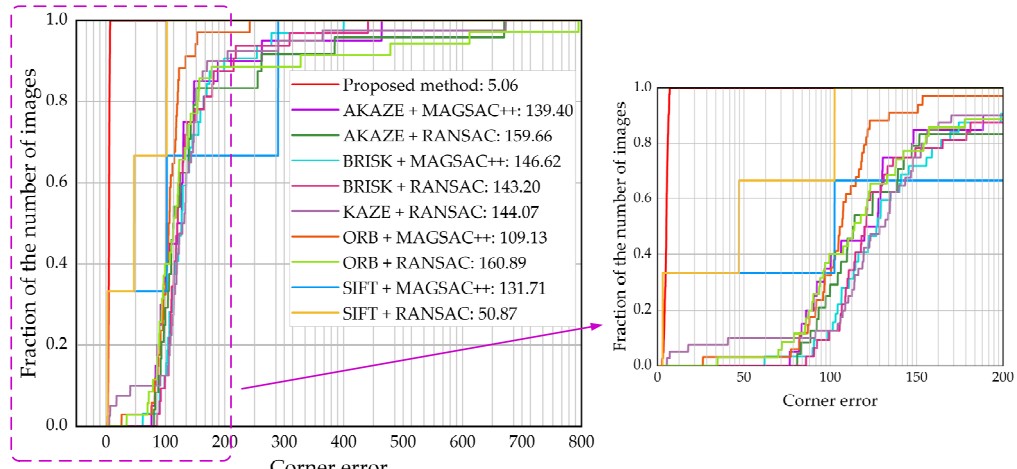

**Figure 10.** Quantitative comparison of HomoMGAN with feature-based methods in the synthetic benchmark dataset. The left side shows the evaluation results of our method and nine feature-based comparisons. The right side represents the local amplification result of the corner error [31] belonging to the range of 0 to 100.

**Quantitative comparison.** Figure 10 shows the quantitative comparison of our method with the feature-based methods in the synthetic benchmark dataset. As shown in Figure 10, the corner error [31] of the proposed HomoMGAN method is much lower than that in the feature-based method, showing superior performance in infrared and visible scenes. In par-

ticular, SIFT [8] + RANSAC [11] and SIFT [8] + MAGSAC++ [13] both show three ladders, which is because they only predict homography matrices on three images. Specifically, if algorithm failure occurs in multiple test images, the fraction of the number of images in a larger range remains unchanged at a certain value of corner error [31] in Figure 10, i.e., a ladder shape appears. Although SIFT [8] + RANSAC [11] fails in most test images in Figure 10, its corner error is the lowest among all of the traditional feature-based methods, confirming the conclusion in the qualitative comparison. In addition, the evaluation values of the remaining eight feature-based methods are close, so we locally zoom in on them, as shown on the right side in Figure 10. We can see that the performance of ORB [10] + MAGSAC++ [13] is the best among these eight methods, but the performance of ORB [10] + RANSAC [11] is the worst. At the same time, it is not difficult to see that the feature-based methods have ladders to varying degrees, i.e., the homography matrix cannot be predicted in some test images, but the proposed HomoMGAN method does not experience algorithm failure.

Figure 11 shows the quantitative comparison of our method with the deep learning-based methods in the synthetic benchmark dataset. As shown in Figure 11, the proposed HomoMGAN method achieves the best performance among deep learning-based methods and significantly outperforms CADHN [15]. According to the trend in the figure, HomoMGAN has a lower corner error [31] than CADHN [15] for most of the test images.

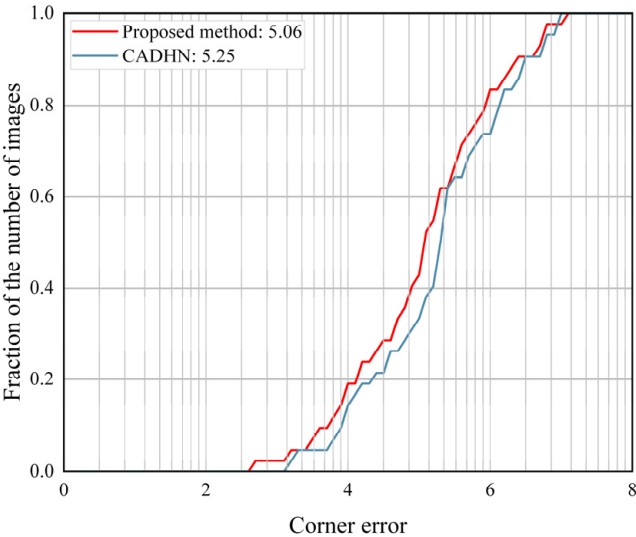

**Figure 11.** Quantitative comparison of HomoMGAN with deep learning-based methods on synthetic benchmark dataset.

**Discussions.** The synthetic benchmark dataset comes from several real-world outdoor scenes, including parking lots, schools, parks, etc. After experimental verification, our method can obtain a more accurate homography matrix in these scenarios, and the registration effect is more reasonable. Since the essence of our synthetic benchmark dataset comes from registered real-world datasets, which to some extent reflect the unregistered situation of real-world data, our method also effectively performs when using unregistered real-world datasets.

### 4.4. Ablation Experiment

**Shallow feature extraction network.** The effectiveness of the shallow feature extraction networks was mainly verified from two perspectives relating to the synthetic benchmark dataset. First, the shallow feature extraction network was replaced with the feature extractor and mask predictor [15], whose feature map comparison is shown in Figure 12. The main reason that this replacement was undertaken is that they function similarly; they both obtain weighted feature maps, highlighting features that are mean-

ingful for homography estimation. As shown in Figure 12, it can be seen that the feature maps produced by our shallow feature extraction network are sparser, which is beneficial in reducing the noise in the feature maps. To prove the effectiveness of our shallow feature extraction network more clearly, we show the corner error [31] of two sample image pairs in Figure 12, as shown in Table 2. It can be seen in Table 2 that the corner error [31] of the shallow feature extraction network on both sets of sample image pairs is much lower than that of the feature extractor and mask predictor [15], proving that the sparse features in our method can effectively reduce the noise in the feature map, improving the accuracy of the homography matrix. In order not to lose generality, we also compare the average corner error [31] on the test set of the feature extractor and mask predictor [15] and shallow feature extraction network; the result is shown in Figure 13. It can be seen that the homography estimation performance of our method is significantly better than that of the feature extractor and mask predictor [15], fully confirming the effectiveness of our shallow feature extraction network. In addition, Table 3 shows the comparison of the number of parameters and computations before and after replacing the shallow feature extraction network, and the results are shown in rows 2 and 3 in Table 3, respectively. Although the number of parameters and computation of the proposed method is slightly higher than the results in row 3, the corner error of the proposed method is significantly lower than that in row 3 (from 5.3 to 5.06).

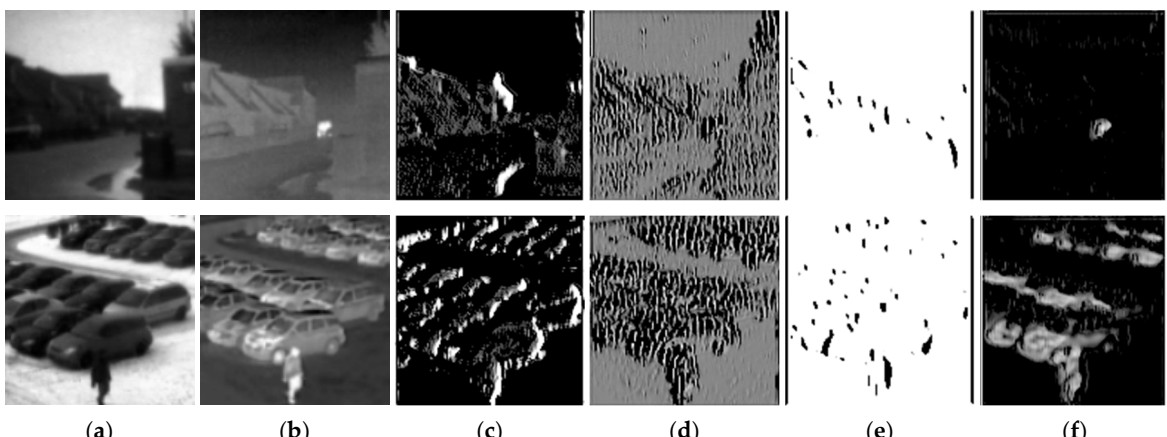

(a)    (b)    (c)    (d)    (e)    (f)

**Figure 12.** Comparison of feature map of feature extractor and mask predictor [15] and shallow feature extraction network. From left to right: (**a**) visible image patch, (**b**) infrared image patch, (**c**) visible fine-feature map in feature extractor and mask predictor [15], (**d**) infrared fine-feature map in feature extractor and mask predictor [15], (**e**) visible fine-feature map in shallow feature extraction network (proposed method), and (**f**) infrared fine-feature map in shallow feature extraction network (proposed method).

**Table 2.** Comparison of corner error [31] of feature extractor and mask predictor [15] and shallow feature extraction network. Rows 2 and 3 represent the corner error [31] corresponding to the two sets of sample images in Figure 12, respectively. Column 2 represents the corner error [31] obtained by replacing the shallow feature extraction network with a feature extractor and mask predictor [15]. Column 3 represents the corner error [31] obtained using a shallow feature extraction network.

| Sample | Feature Extractor and Mask Predictor [15] | Shallow Feature Extraction Network |
|---|---|---|
| (1) | 2.96 | 2.69 |
| (2) | 4.43 | 4.11 |

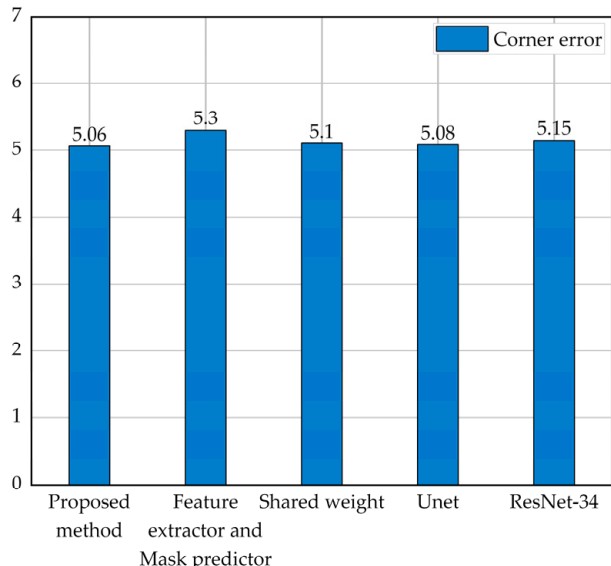

**Figure 13.** Ablation experiments. Feature extractor and mask predictor [15] and w/o. Shared weight are ablation experiments for shallow feature extraction networks. Unet [38] and ResNet-34 [39] are ablation experiments on generator backbones.

**Table 3.** Comparison of the number of parameters and computations under different ablation settings.

| Settings | Parameters (G) | Computations (MB) |
|---|---|---|
| Proposed method | 12.71 | 70.58 |
| Feature extractor and Mask predictor | 12.47 | 70.57 |
| Shared weight | 12.85 | 70.56 |
| Unet | 27.71 | 34.09 |
| ResNet-34 | 4.45 | 24.79 |

Second, the two conditions of the shallow feature extraction network were compared—"shared weight" and "w/o. shared weight" (the proposed method)—and the fine-feature map results are shown in Figure 14. We show the output of the shallow feature extraction network to analyze the impact of "shared weight" and "w/o. shared weight" on the shallow feature extraction network. As shown in Figure 14c,d, under the condition of shared weight, the fine-feature map output by the shallow feature extraction network retains a large number of dense features. However, this introduces significant noise into the homography estimation process, degrading homography estimation performance. Interestingly, after we no longer share the weight for the shallow feature extraction network, the network can extract features according to the imaging characteristics of each type of image, thereby reducing the impact of noise in homography estimation, as shown in Figure 14e,f. In the case of "w/o. shared weight", both the infrared image and the visible image not only retain meaningful features for homography estimation, but the number of features is significantly less than that under the "shared weight" conditions. The corner error [31] in Table 4 also demonstrates the advantage of "w/o. shared weight"; the corner error [31] of "w/o. shared weight" is significantly lower than that of "shared weight" in these two sets of images. Similarly, as seen in Figure 13, "w/o. shared weight" can significantly reduce the error from 5.1 to 5.06 compared with "shared weight", which also proves the effectiveness of the shallow feature extraction network in our method without shared weight. In addition, Table 3 compares the number of parameters and computations of "w/o. shared weight" and "shared weight", as shown in rows 2 and 4 in Table 3. The parameter amount in row 2 is slightly lower than in row 4, but its computation amount is slightly higher than in row 4.

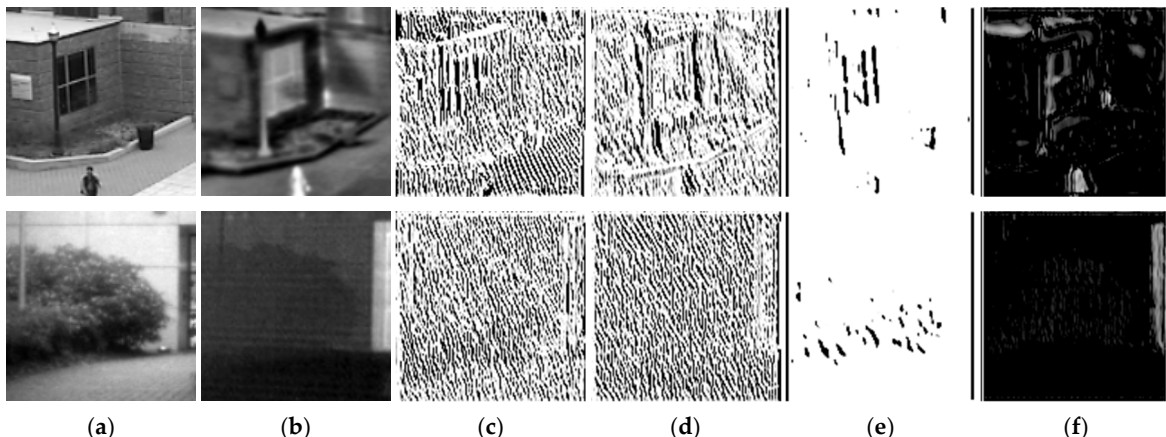

|     |     |     |     |     |     |
| :-: | :-: | :-: | :-: | :-: | :-: |
| (**a**) | (**b**) | (**c**) | (**d**) | (**e**) | (**f**) |

**Figure 14.** Comparison of fine-feature maps for shallow feature extraction network with and without shared weight. From left to right: (**a**) visible image patch, (**b**) infrared image patch, (**c**) visible fine-feature map in shared weight, (**d**) infrared fine-feature map in shared weight, (**e**) visible fine-feature map in w/o. shared weight (proposed method), and (**f**) infrared fine-feature map w/o. shared weight (proposed method).

**Table 4.** Comparison of corner error [31] for shallow feature extraction network with and without weight sharing. Row 2 and 3 represent the corner error [31] corresponding to the two sets of sample images in Figure 14, respectively. Column 2 represents the corner error [31] resulting from shallow feature extraction network in shared weight. Column 3 represents the corner error [31] resulting from shallow feature extraction network w/o. shared weight.

| Sample | Shared Weight | w/o. Shared Weight |
| :-: | :-: | :-: |
| (1) | 6.10 | 5.99 |
| (2) | 3.23 | 3.12 |

**Generator backbone.** Several common backbones were investigated, including Unet [38] and ResNet-34 [39], to verify the effectiveness of the generator backbone in the proposed method. As shown in Figure 12, our method achieves the best performance among these several backbones and slightly outperforms Unet [38]. The homography estimation generator of the proposed method and Unet [38] use the encoder–decoder structure to fuse shallow low-level features in the deep network, and the corner error of both is better than ResNet-34 [39]. This fully illustrates that shallow low-level features can effectively improve homography estimation performance. Interestingly, Unet [38] can be used to directly predict the homography matrix, and the performance is significantly better than the common backbone (ResNet-34 [39]) in typical homography estimation methods [15,16]; the corner error [31] dropped significantly from 5.15 to 5.08. To the best of our knowledge, Unet [38] has not been used to directly predict homography matrices before. In addition, Table 3 shows the comparison of the parameter amount and computation amount before and after generator backbone replacement, as shown in rows 2, row 5, and row 6 in Table 3. Although ResNet-34 [39] has the smallest number of parameters and computations, it has the highest corner error compared to the backbones.

## 5. Conclusions

For infrared and visible scenarios, we proposed an end-to-end unsupervised homography estimation method (HomoMGAN). In HomoMGAN, we designed a shallow feature extraction network with unshared weights to extract fine meaningful features for homography estimation from infrared and visible images so that the network focuses on important features of the source image to improve homography estimation performance. In addition, we also designed an unsupervised GAN to predict the homography matrix directly and

forced the fine features in the warped image to be closer to the fine features of the target image by an adversarial game between the generator and the discriminator. We demonstrated the superiority and effectiveness of HomoMGAN through qualitative and quantitative comparisons with ten other methods on a synthetic benchmark dataset. Notwithstanding, our method has its limitations, such as limited homography estimation performance in low-light scenes. Therefore, we will further optimize HomoMGAN in future work so that it can be better applied to homography estimation tasks in low-light scenes.

**Author Contributions:** Conceptualization, Y.L. and X.W.; methodology, Y.W.; software, C.S.; validation, X.W., Y.W. and C.S.; formal analysis, Y.L.; investigation, Y.W.; writing—original draft preparation, Y.L. and X.W.; writing—review and editing, Y.L., X.W., Y.W. and C.S.; project administration, Y.L.; funding acquisition, Y.L and Y.W. All authors have read and agreed to the published version of the manuscript.

**Funding:** This research was funded in part by the National Key R&D Program of China (program no. 2021YFF0603904) and in part by the Fundamental Research Funds for the Central Universities (program no. ZJ2022-004, and no. ZHMH2022-006).

**Institutional Review Board Statement:** Not applicable.

**Informed Consent Statement:** Not applicable.

**Data Availability Statement:** Not applicable.

**Acknowledgments:** We sincerely thank the authors of SIFT, ORB, KAZE, BRISK, AKAZE, and CADHN for providing their algorithm codes to facilitate the comparative experiment. Meanwhile, we would like to thank the anonymous reviewers for their valuable suggestions, which were of great help in improving the quality of this paper.

**Conflicts of Interest:** The authors declare no conflict of interest.

## Abbreviations

The following abbreviations are used in this manuscript:

| | |
|---|---|
| HomoMGAN | Homography estimation method using Multi-scale Generative Adversarial Network |
| GAN | Generative Adversarial Network |
| SIFT | Scale Invariant Feature Transform |
| SURF | Speeded Up Robust Features |
| ORB | Oriented FAST and Rotated BRIEF |
| BRISK | Binary Robust Invariant Scalable Keypoints |
| AKAZE | Accelerated-KAZE |
| LPM | Locality Preserving Matching |
| GMS | Grid-Based Motion Statistics |
| BEBLID | Boosted Efficient Binary Local Image Descriptor |
| LIFT | Learned Invariant Feature Transform |
| SOSNet | Second-Order Similarity Network |
| OAN | Order-Aware Networks |
| RANSAC | Random Sample Consensus |
| MAGSAC | Marginalizing Sample Consensus |
| ALLSSA | Antileakage LSSA |
| DLT | Direct Linear Transformation |
| VGG | Visual Geometry Group |
| STN | Spatial Transformation Network |
| CBAM | Convolutional Block Attention Module |
| ReLu | Rectified Linear Unit |
| ResNet-34 | 34-layer deep Residual Network |

## Appendix A. Dependency on $\lambda$, $\mu$, and $a$ and $b$

In Table A1, the parameters in the loss function on the synthetic benchmark dataset are analyzed, showing the corner error they obtained for different values to demonstrate our

fine-tuning process. For parameter $\lambda$, the homography loss (corresponding to the weight $\lambda$) accounts for an increase, which significantly affects the homography estimation performance. For parameter $\mu$, reducing the proportion of the adversarial loss (corresponding to the weight $\mu$) reduces the adversarial effect of the GAN, thereby reducing homography estimation performance. It is worth noting that parameter $a$ represents the label value of the fine-feature map of the target image, and parameter $b$ represents the label value of the fine-feature map of the warped image. $a$ and $b$ are not specific values, but so-called soft labels. At the same time, the discriminator in the proposed method uses the Sigmoid function to predict the probability value, so the discriminator's prediction probability for the fine-feature map of the target image does not exceed one, and the prediction probability for the fine-feature map of the warped image is not lower than zero. For parameters $a$ and $b$, when $a$ is set to a random number from 0.95 to 1 and $b$ is set to a random number from 0 to 0.05, the corner error is the lowest.

**Table A1.** Dependency on $\lambda$, $\mu$, and $a$ and $b$. We demonstrate the corner error for different values of parameters using a synthetic benchmark dataset.

| $\lambda$ | Corner Error |
| --- | --- |
| 0.05 | 5.21 |
| 0.01 | 5.06 |
| 0.005 | 5.09 |

(a) dependency on $\lambda$ with $\mu = 0.005$, $a \in [0.95, 1]$, and $b \in [0, 0.05]$

| $\mu$ | Corner Error |
| --- | --- |
| 0.01 | 5.09 |
| 0.005 | 5.06 |
| 0.001 | 5.40 |

(b) dependency on $\mu$ with $\lambda = 0.01$, $a \in [0.95, 1]$, and $b \in [0, 0.05]$

| $a$ and $b$ | Corner Error |
| --- | --- |
| $a \in [0.98, 1]$, $b \in [0, 0.02]$ | 5.08 |
| $a \in [0.95, 1]$, $b \in [0, 0.05]$ | 5.06 |
| $a \in [0.8, 1]$, $b \in [0, 0.2]$ | 5.23 |

(c) dependency on $a$ and $b$ with $\lambda = 0.01$ and $\mu = 0.005$

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
