# Peer review of "Infrared and Visible Image Homography Estimation Using Multiscale Generative Adversarial Network"

_electronics, doi:10.3390/electronics12040788_

Round 1

Reviewer 1 Report

This paper is well written and presents a homography for infrared images. Some comments are:

1) The literature can be improved

Makantasis, Konstantinos, et al. "Data-driven background subtraction algorithm for in-camera acceleration in thermal imagery." IEEE Transactions on Circuits and Systems for Video Technology 28.9 (2017): 2090-2104.

Luo, Yinhui, et al. "Detail-Aware Deep Homography Estimation for Infrared and Visible Image." Electronics 11.24 (2022): 4185.

Debaque, Benoit, et al. "Thermal and visible image registration using deep homography." 2022 25th International Conference on Information Fusion (FUSION). IEEE, 2022.

2) the computational cost should be given

3) the effect of different parameters should be given 

4) The main limitation of the paper is the lack of real-word images. Please discuss at least what will happen? 

Reviewer 2 Report

This paper deals with an infrared and visible image homography estimation using multiscale generative adversarial network. I would like to point out following.

1.    What is main objective of this paper, it is not clear in introduction even though authors mentioned from line 64 to 82 in introduction.

2.    What is the main differencing for the previous work and proposed method?

3.    What’s the difference or novelty.

Author Response

请参阅附件。

Reviewer 3 Report

Reviewer’s Report on the manuscript entitled:

Infrared and Visible Image Homography Estimation Using Multiscale Generative Adversarial Network

The authors designed an infrared and visible image Homography estimation method using Multi-scale Generative Adversarial Network for feature extraction. They also proposed an unsupervised generative adversarial network to predict the homography matrix directly. The topic, method, and results are interesting and sound. I have a few suggestions for further improvement as listed below.

Lines 9 and 10. Grammar issue. Please rewrite the sentence.

Line 36. Please remove “usually”

Lines 118-120. All the abbreviations must be defined the first time they appear. If you do not use the abbreviation later in the text, then please just provide the full name. Line 129. Define VGG, etc. Please also add an acronym table at the end of the manuscript listing all the abbreviations used in the manuscript.

In the Related Work, please add the following recent article which compares ResNET with VGG for image classification:

https://doi.org/10.3390/s21238083

Line 122. Reference 27 is old. Please replace them with more recent ones. For example, you can add the ALLSSA method which is a robust regression method based on spectral analysis which prevents over/under fitting issues:

https://doi.org/10.1190/geo2017-0284.1

Line 258. Did you also use techniques such as early stopping to speed up the computation and mitigate the over fitting issue as described in the first article I suggested above? Please elaborate on it here.

The limitations of this study should be mentioned in the Conclusion. Please also mention the computational complexity of HomoMGAN. Please also use past verbs in the Conclusion section.

Please carefully proofread the manuscript.

Round 2

Reviewer 1 Report

All are OK now.